# LATENT ACTION ROBOT FOUNDATION WORLD MODELS FOR CROSS-EMBODIMENT ADAPTATION

## ABSTRACT

Robot action-conditioned video generation models, also known as robot world models, hold great potential for enhancing robotic planning and decision-making. However, the diversity of robot embodiments and action spaces makes it challenging to build models that generalize across different embodiments. We introduce a robot foundation world model, Latent Action Conditioned World Model (LAC-WM), which operates within a learned unified latent action space shared across diverse embodiments. We explore how this unified action space improves the world model's performance when adapted to previously unseen robot embodiments. Specifically, we compare LAC-WM to a baseline model, Explicit Action Conditioned World Model (EAC-WM) conditioned on explicit motion labels. Our results show that conditioning on explicit labels creates disjoint action spaces across embodiments, limiting downstream task performance when adapting to new robots. We evaluate both models on a dexterous manipulation task. The latent action-conditioned model LAC-WM achieves up to a 46.7% improvement in performance over EAC-WM. Crucially, the unified latent action space allows LAC-WM's downstream performance to scale positively with the number of embodiments used during pretraining. In contrast, the disjoint action space in EAC-WM leads to decreased performance as the number of pretraining embodiments increases. These results highlights the importance of a unified action space for efficient cross-embodiment learning, addressing a key challenge in robotics.

## 1 INTRODUCTION

Robot action-conditioned video generation models, or *robot world models*, offer a complementary approach to robot policies based on imitation learning, allowing robots to predict future states and evaluate planned actions before execution, improving planning and decision-making. However, a fundamental and persistent challenge in robotics is the vast and evolving diversity of robot embodiments. Robots differ widely in their physical structure, sensors and actuators, and control strategies, and new embodiments are constantly emerging. This creates a critical need for robot world models that can efficiently learn from cross-embodiment data Zheng et al. (2025a); Liu et al. (2025); Zheng et al. (2025b); Doshi et al. (2024); Bu et al. (2025b) and adapt quickly to previously unseen robot forms.

To address the need, we propose learning a unified latent action space that aligns action representations across different embodiments, including both human and different robot agents. This latent space abstracts away embodiment-specific details while preserving task-relevant information, enabling the model to integrate heterogeneous data during pretraining. We show that such a unified latent space will empower the world model to learn more efficiently across embodiments during pretraining and adapt quickly to unseen embodiments through finetuning, laying the groundwork for a truly generalizable foundational world model for robotics.

Latent action spaces have primarily been explored for extracting action representations from unlabeled video data (Ye et al., 2024; Chen et al., 2024). However, most robotic datasets come with action annotations, and advances in human pose estimation provide scalable motion information, including detailed hand poses and camera movements. This enables learning action spaces directly from motion data. Despite this, there is a lack of comparative analysis for action embedding alignment across different embodiments between representations derived directly from motion data and those learned through latent action models. We hypothesize that training action encoders solely on motion data

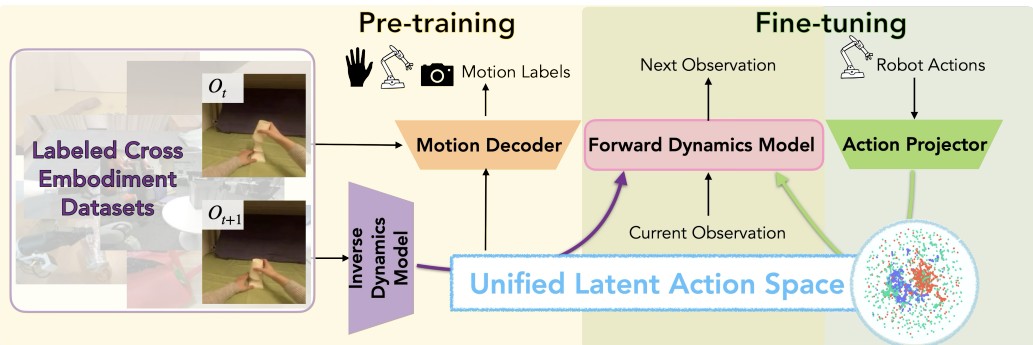

Figure 1: **LAC-WM Model Architecture**. During pretraining (yellow area), the inverse dynamics model (IDM) encodes image observations from various embodiments into a unified latent action space, as the condition for the forward dynamics model (FDM). A motion decoder translates the latent action space into motion labels, including delta human hand poses, robot end-effector actions, and camera motion. During finetuning (green area), an action projector encodes raw robot actions of previously unseen embodiments into the unified latent action space, enabling the world model to condition on raw robot actions.

leads to disjoint latent spaces for different embodiments (Figure 2), which limits cross-embodiment knowledge transfer and degrades generalization performance on unseen embodiments.

We focus on learning a foundational robot world model that learns efficiently from cross-embodiment data during pretraining and adapts quickly to unseen embodiments through finetuning. Specifically, we investigate two key research questions: (1) *Do latent action models better unify action representations across diverse embodiments compared to directly using action labels?* and (2) *Does conditioning a world model on a unified action space enhance cross-embodiment learning and downstream robot task planning performance compared to a disjoint latent space?*

We propose the Latent Action Conditioned Robot World Model (LAC-WM), which predicts future observations conditioned on a unified latent action space. LAC-WM consists of four components: an inverse dynamics model (IDM), an forward dynamics model (FDM), a motion decoder and an action projector, as in Figure 1. The IDM encodes pairs of consecutive image observations into latent actions, which then condition the FDM to predict the next frame. Additionally, we incorporate an auxiliary loss, following Nikulin et al. (2025), to decode latent actions back into explicit motion labels using the motion decoder. The IDM and FDM are trained jointly on datasets combining human manipulation videos and robot data. Since future observations, required by the IDM, are unavailable at inference time, we train an action projector that maps explicit actions into the latent action space (Gao et al., 2025), enabling the world model to directly consume action inputs.

We evaluate LAC-WM in a cross-embodiment transfer setting, where the model is pretrained on multiple embodiments and then fine-tuned on a previously unseen robot using the action projector. We benchmark against an Explicit Action Conditioned World Model (EAC-WM), which encodes motion information directly into action latents. Our experiments demonstrate that LAC-WM generates higher-quality future predictions compared to the explicit action baseline (EAC-WM). Furthermore, when applied to action selection for a dexterous manipulation task, LAC-WM achieves up to a 46.7% improvement in task success rate over EAC-WM. Notably, we observe a positive scaling relationship between downstream planning performance and the number of embodiments used during pretraining for LAC-WM, whereas EAC-WM exhibits the opposite trend, likely due to its disjoint latent space, which impedes efficient cross-embodiment learning.

## 2 RELATED WORK

**World Models in Robotics.** World models (WMs) have emerged as a way to achieve controllable video generation (Bruce et al., 2024; Zhang et al., 2025b; Li et al., 2025; Po et al., 2025; Yang et al., 2023; Valevski et al., 2024; Yu et al., 2025). Robotic world models serve as scalable and explainable compliments to imitation learning based policies. Prior work has explored training WMs for navigation (Hu et al., 2023; Bar et al., 2025; Gao et al., 2024; Hassan et al., 2025) with well-defined action spaces (e.g., velocity). WMs for robot manipulation involve diverse action spaces due to differences in robot morphologies, and as a result, they typically rely on action-free video prediction during pretraining (Gupta et al., 2022; Wu et al., 2024; Assran et al., 2025), or use text or

video conditioning (Agarwal et al., 2025), adding action conditioning only during fine-tuning (Gupta et al., 2022; Wu et al., 2024; Agarwal et al., 2025; Assran et al., 2025). Zhu et al. (2025) incorporate action conditioning for a single robot embodiment but do not evaluate cross-embodiment transfer. In contrast, LAC-WM directly tackles the challenge of heterogeneous action spaces with a unified latent action space, enabling efficient cross-embodiment learning and better generalization.

**Latent Action Modeling.** Latent Action Models (LAM) have emerged as a promising approach to learn action representations from large-scale unlabeled data. Prior work has explored LAMs for training *latent action* policies for game environments (Schmidt & Jiang, 2024), and fine-tuning expert-level policies from these, with limited labels. Prior work have explored LAMs for robot manipulation, by extracting latent actions from large-scale unlabeled human videos as labels to fine-tune vision-language models, and subsequently vision-language-action (VLA) models (Ye et al., 2024; Chen et al., 2025; Bu et al., 2025b; Jang et al., 2025; Cui et al., 2024; Kim et al., 2025). Liang et al. (2025); Nikulin et al. (2025) introduced an auxiliary motion decoding loss which aligns latent actions to explicit control actions, to encourage the latent space to capture actionable content. Zhang et al. (2025a) explores various *best practices* for training LAMs. LAC-WM also uses a motion decoding loss and learns a continuous latent action space. Instead of focusing on policy learning as in prior work, we learn an aligned latent action space across labeled datasets from different embodiments, for building robot world foundation models.

**Latent Action World Models.** Bruce et al. (2024); Ren et al. (2025) introduce a latent action model (LAM) paired with a world model, trained on datasets that include robot data. However, their approach does not involve cross-embodiment pretraining across diverse embodiments, nor does the world model support conditioning on raw robot actions at inference time. Chen et al. (2024) train LAMs with data augmentation to regularize latent actions that feed into a foundation policy and world models for robotic manipulation. However, they focus on action-free video pretraining and do not demonstrate adaptation to unseen robot embodiments. Gao et al. (2025) demonstrate that a pretrained latent action-conditioned world model can be adapted for robot manipulation by projecting raw action signals into the latent action space. However, they focus on action-free video pretraining and lack comparisons with robot world models trained using explicit action labels. In contrast, we leverages labeled cross-embodiment data for robot world model training and investigate the comparison between a learned latent action space and directly encoding the motion labels. We show a unified latent action space benefits cross-embodiment learning with better downstream robot planning performance, particularly when adapting to unseen embodiments.

## 3 METHOD

We propose LAC-WM, a robot foundation world model conditioned on latent actions. LAC-WM is pretrained on cross-embodiment datasets and finetuned to adapt to previously unseen robot embodiments. The overall framework is illustrated in Figure 1. During pretraining (left, yellow), LAC-WM consists of three components: an inverse dynamics model (IDM), a forward dynamics model (FDM), and a motion decoder (MD). The IDM encodes multi-embodiment image observations into a unified latent action space, which conditions the FDM to predict the next image latent. An auxiliary loss from the MD decodes latent actions into explicit motion labels. Both the image prediction and motion decoding losses jointly supervise all modules. During finetuning (right, green), an action projector maps raw robot actions of an unseen robot embodiment into the latent action space, enabling the pretrained FDM to generate rollouts conditioned on embodiment-specific actions.

### 3.1 LATENT ACTION CONDITIONED ROBOT WORLD MODEL

Given a video demonstration with $T$ frames $\{O_1, ..., O_T\}$ and corresponding motion information $\{a_1, ..., a_T\}$, LAC-WM works as follows.

**Latent Action Modeling with an IDM and FDM** First, RGB observations $O_t$ are mapped into visual embeddings $x_t$ using a visual encoder (V-JEPA-2 (Assran et al., 2025)). Next, the IDM takes as input two consecutive embeddings $x_t$ and $x_{t+1}$ to encode latent action $z_t$ such that $z_t =$ IDM$(x_t, x_{t+1})$. The IDM uses $M$ causal attention layers to produce contextualized representations of $x_t$ and $x_{t+1}$, followed by $N$ cross-attention layers that transform a learned query token $q_t$ into latent actions $z_t$ by cross-attending to the contextualized representations. The FDM processes $x_t$ and

$z_t$ using $L$ blocks composed of self-attention on $x_t$, self-attention on $z_t$, and cross-attention from $x_t$ to $z_t$, to predict the next visual embedding $\hat{x}_{t+1}$, such that $\hat{x}_{t+1} = \text{FDM}(x_t, z_t)$.

**Cross-Augmentation Inputs**  Since the IDM is partially supervised by the image latent prediction loss, it may encode future information directly into $z_t$ as a shortcut. To mitigate this, we adopt a cross-augmentation training scheme, similar to Chen et al. (2024). Given observations $O_t$ and $O_{t+1}$, we apply independent augmentations $\mathcal{A}_1$ and $\mathcal{A}_2$ to obtain two augmented embedding pairs: $(x_t^1, x_{t+1}^1)$ and $(x_t^2, x_{t+1}^2)$. One pair is used by IDM to generate $z_t^1 = \text{IDM}(x_t^1, x_{t+1}^1)$, while the other is used by FDM to predict $\hat{x}_{t+1}^2 = \text{FDM}(x_t^2, z_t^1)$. The image latent prediction loss is given by:

$$\mathcal{L}_{\text{recon}} = \|\hat{x}_{t+1}^2 - x_{t+1}^2\|_2^2,$$

**Motion Decoder**  While prior work mitigates shortcut solutions in LAM training by using quantization or low-dimensional latent spaces (Chen et al., 2024; Ye et al., 2024), such constraints limit representational capacity. Instead, LAC-WM uses continuous latent actions and mitigates shortcuts through an auxiliary motion decoding loss (Nikulin et al., 2025; Liang et al., 2025).

The motion decoder (MD) reconstructs ground-truth motions $\mathbf{a}_t$ from latent actions $z_t$, conditioned on the current observation to enable viewpoint-dependent predictions. Specifically, $z_t$ serves as a query in a cross-attention module over visual tokens extracted from the current frame. To reduce computation, the visual tokens are downsampled via a 2D convolutional layer. The resulting cross-attended features are fed into an MLP to produce motion outputs, such that $\hat{\mathbf{a}}_t = \text{MD}(x_t, z_t)$. The decoder is trained using a mean squared error loss:

$$\mathcal{L}_{\text{motion}} = \|\hat{\mathbf{a}}_t - \mathbf{a}_t\|_2^2.$$

The overall training loss combines the self-supervised image latent prediction loss and supervised motion decoding loss using weighting coefficients $\lambda_{\text{recon}}$ and $\lambda_{\text{motion}}$ as follows:

$$\mathcal{L} = \lambda_{\text{recon}}\mathcal{L}_{\text{recon}} + \lambda_{\text{motion}}\mathcal{L}_{\text{motion}}.$$

The IDM, FDM, and MD are trained end-to-end. Once trained, we keep the FDM and add an action projector for finetuning to adapt to an unseen embodiment.

### 3.2 ADAPTING TO UNSEEN ROBOT EMBODIMENTS

The latent action model requires future observations, which are unavailable during inference. Prior works address this by training a policy conditioned on current observations to generate latent actions (Ye et al., 2024; Chen et al., 2024). However, this approach does not support planning with a raw-action-conditioned world model. Instead, we learn an *action projector* (Figure 1, right) that maps explicit raw actions into the latent action space (Gao et al., 2025). This projector is a two-layer MLP that takes the action $\mathbf{a}_t$ as input and outputs the corresponding latent action $z_t$. To adapt to an unseen robot embodiment, we initialize an action projector and finetune the pre-trained LAC-WM in three stages. First, we fine-tune the IDM and FDM of LAC-WM end-to-end using LoRA (Hu et al., 2021) with rank 2. Second, we freeze the FDM and train the action projector from scratch to map explicit actions into the latent space. Third, we jointly fine-tune the projector and FDM end-to-end using LoRA with rank 2. At inference time, we use only the action projector and FDM to perform action-conditioned imagined rollouts.

## 4 EXPERIMENTS

### 4.1 BASELINE

We use a robot world model with explicit action encoders as our baseline. Specifically, we adopt the same FDM architecture but generate action conditioning by directly encoding explicit motions, such as robot end-effector (EE) actions or delta human hand poses—into action embeddings. Since different embodiments have distinct action spaces, we employ separate action encoders for each embodiment, following common practice in multi-embodiment learning (NVIDIA et al., 2025). Each action encoder shares the same architecture as the action projector in LAC-WM, taking only the

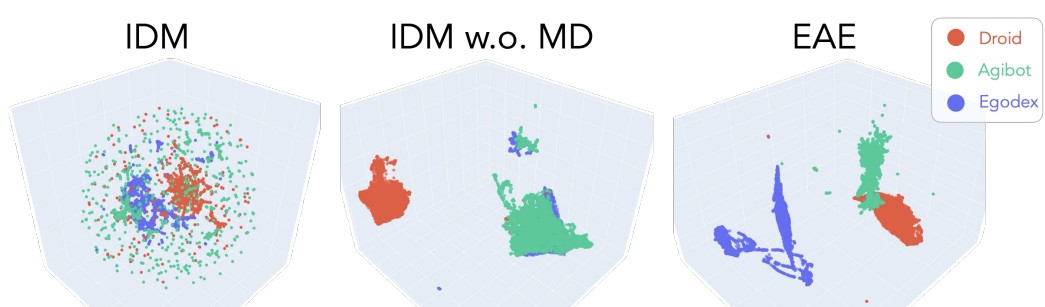

**Figure 2: UMAP visualization of 7,000 action embeddings** from three datasets: Droid (red), Agibot (green), and Egodex (blue). **Left**: Embeddings from the IDM, showing strong cross-dataset alignment. **Middle**: IDM trained without the motion decoder (MD), where Agibot and Egodex cluster together but separate from Droid. **Right**: Explicit Action Encoder (EAE) embeddings, which are distinctly separated by dataset.

explicit action as input and outputting the corresponding action embedding. To adapt to unseen embodiments, we train a new action encoder from scratch and fine-tune the FDM using LoRA with rank 2. We denote this model, consisting of explicit action encoders and the FDM, as the explicit action conditioned robot world model (EAC-WM).

## 4.2 MODEL PRETRAINING

**Datasets** We conduct pretraining on three datasets: EgoDex (Hoque et al., 2025), Agibot (Bu et al., 2025a), and Droid (Khazatsky et al., 2025). Agibot features a bimanual humanoid robot and Droid consists of a single Franka arm with a Robotiq gripper. EgoDex is a dataset of humans manipulating different objects in a tabletop environment. The datasets provide camera extrinsic information corresponding to egocentric head cameras for EgoDex and Agibot and third-person view cameras for Droid. Additionally, we have robot end-effector action labels for Agibot and Droid and human hand poses annotations for EgoDex. For each dataset, we chunk the actions into 5-step sequences to down-sample the observation frequency, which we found in practice improves world model learning. We sample 50,000 trajectories from each dataset. The action space includes changes in end-effector poses and camera poses. More details about the dataset action space is in Section A.2.

**Parameters** Both EAC-WM and LAC-WM are trained for 80,000 iterations with a batch size of 512. Both models employ an action embedding dimension of 64. For both models, we use a pretrained V-JEPA2 (Assran et al., 2025) RGB tokenizer for image encoding, which is frozen during the model pre-training and fine-tuning. We use a custom V-JEPA2 RGB decoder to decode the predicted image embeddings into RGB image space as in Section A.4. More details are in Section A.5.

## 4.3 MODEL FINETUNING

As described in Section 3.2, we finetune models to adapt to an unseen embodiment. Specifically, we use a bimanual Franka robot setup equipped with Allegro hands, which we refer to as BFA.

**Dataset** We consider a pick-and-place task that involves lifting an object from a kitchen counter and dropping it back down. The training dataset consists of 7,265 trajectories collected in the RoboCasa simulator (Nasiriany et al., 2024), covering 22 object categories (e.g., apple, bell pepper, ketchup), and includes both successful and failed attempts. The trajectories only use the right robot arm. Thus, the action space is 25D, consisting 16 joint dimensions for the robot-hand, 3 dimensions for end-effector position, and 6 dimensions for rotation. Egocentric visual observations are captured using a head camera. As in Section 4.2, actions are chunked into 5-step sequences, reducing the observation frequency. More details are in Section A.3.

**Finetuning Parameters** We finetune the LAC-WM and EAC-WM models on the BFA dataset using a batch size of 256 for a total of 60k iterations. For LAC-WM, fine-tuning follows the multi-stage procedure outlined in Section 3.2, with 20k iterations for Stage 1, 5k for Stage 2, and 35k for Stage 3. For EAC-WM, we follow the procedure in Section 4.1 and fine-tune for 60k iterations in a single stage. We found that directly fine-tuning EAC-WM in one stage yields better downstream performance than using the multi-stage approach, as shown in Section A.6.

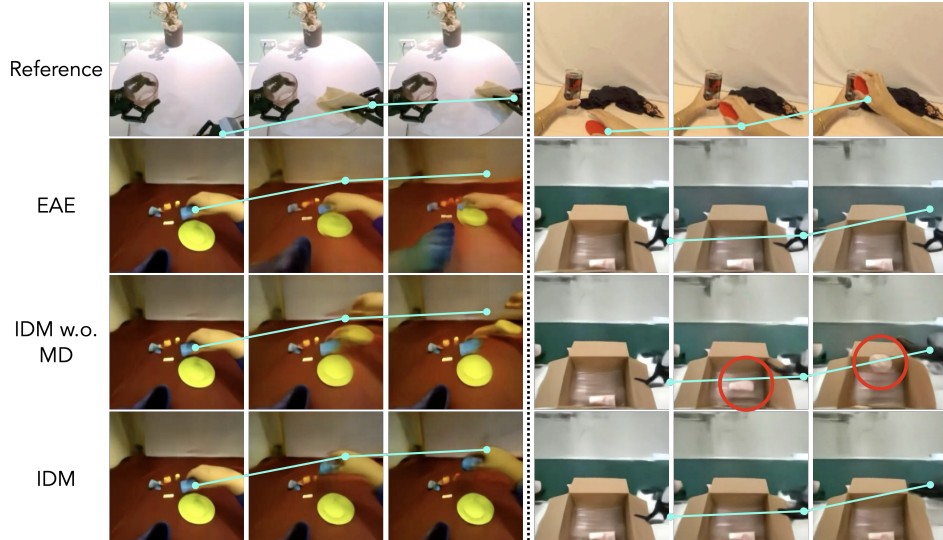

**Figure 3: Cross-embodiment action transfer results**: Left panels depict robot-to-human transfer; right panels show human-to-robot transfer. Row 1 shows reference videos. Rows 2–4 show FDM-generated videos using embeddings from EAE, IDM without MD, and IDM, respectively. Blue dotted lines trace end-effector motion from the references, consistent across generated videos except for start positions. IDM embeddings produce trajectories closely matching references, while others show inaccurate motion or artifacts (red circles).

## 5 LATENT ACTION ANALYSIS

This section addresses the first research question of *whether latent action models better unify action representations across diverse embodiments compared to directly using action labels on heterogeneous datasets*. Specifically, we visualize the latent action space and qualitatively evaluate the action transfer between different embodiments.

**UMAP Visualization** We analyze the action embeddings generated by both the inverse dynamics model (IDM) in the pretrained LAC-WM and the explicit action encoder (EAE) in pretrained EAC-WM by sampling 7,000 data points from the validation splits of the three pre-training datasets. Visualized with UMAP (Figure 2), the latent action embeddings produced by the IDM cluster closely across all three datasets, suggesting they occupy a unified action space (left panel). In contrast, the embeddings from the EAE are clearly separated by dataset, indicating that explicit actions do not share a common latent representation (right panel). Interestingly, when the IDM is trained without the motion decoder, the embeddings from Agibot and Egodex form a joint cluster, distinct from that of Droid (middle panel). We hypothesize this occurs because both Agibot and Egodex are based on egocentric views, resulting in more similar visual distributions. Extracting latent actions purely from visual observations may lack the task-relevant grounding provided by motion data, potentially resulting in latent representations that encode primarily perceptual features. This can lead to disjoint latent action spaces when faced with varying visual distributions.

**Action Latent Transfer** We demonstrate the transferability of latent action embeddings across different embodiments by conditioning the FDM on observations from one embodiment while using action embeddings derived from another embodiment. Qualitative results in Figure 3 show transfers between robot (Agibot) and human motions. Specifically, given an 8-frame reference video, we extract latent action embeddings by passing the video frames through the IDM. For comparison, the explicit actions are encoded using the explicit action encoders (EAE) from EAC-WM, producing explicit action embeddings.

As shown in Figure 3, the learned latent action space enables video generation that captures semantically consistent and similar movements across different embodiments. Specifically, the latent action encodes the end-effector motion direction in image space. As illustrated in the left column of Figure 3, the same latent action prompts both the robot's right end-effector and the human's right hand to move rightward and upward. In contrast, EAE embeddings fail to transfer effectively, producing nearly static poses of human hands and robot grippers. This aligns with our UMAP visualization, where separated action embeddings from different embodiments lead to out-of-distribution conditioning when mixing embodiments. Videos generated with IDM embeddings trained without the MD exhibit

| Model | Unseen Instances | | | | Unseen Categories | | | |
|---|---|---|---|---|---|---|---|---|
| | PSNR↑ | LPIPS↓ | FID↓ | FVD↓ | PSNR↑ | LPIPS↓ | FID↓ | FVD↓ |
| EAC-WM-S | 22.212 | 0.077 | 10.648 | 42.257 | 22.062 | 0.081 | 10.642 | 40.477 |
| EAC-WM | 24.252 | 0.057 | 9.312 | 30.349 | 24.091 | 0.060 | 9.144 | 29.702 |
| LAC-WM | **27.484** | **0.037** | **8.439** | **23.213** | **27.333** | **0.040** | **8.509** | **24.456** |

**Table 1: Action conditioned imagined roll-out performance**. Image and video quality metrics are reported across evaluation splits that test generalization to unseen instances (`Unseen Instance`) and held-out categories (`Unseen Categories`). LAC-WM, which leverages cross-embodiment pretraining with latent action embeddings, achieves the best roll-out performance compared to models trained from scratch (EAC-WM-S) or pretrained with explicit actions (EAC-WM). Best results per split and metric are shown in bold.

artifacts like duplicated hands and floating objects (red circles). These artifacts suggest that the latent embeddings may have learned the shortcut solution of encoding perceptual features such as optical flow when not grounded by motion labels. Conversely, videos generated with action embeddings from IDM trained with MD contain no residual artifacts from the source embodiment, indicating that these embeddings effectively capture embodiment-invariant motion representations.

To summarize, the UMAP visualization and action transfer results suggest that *latent action models do better unify action representations across diverse embodiments than directly using action labels on heterogeneous datasets*. Moreover, incorporating a motion decoding loss further enhances the coherence of the latent action space.

## 6 ROBOT TASK EVALUATION

This section studies the world model performance when adapting to a previously unseen embodiment.

### 6.1 ACTION-CONDITIONED IMAGINED ROLL-OUT

We quantitatively evaluate action conditioned image generation performance of the world models by predicting eight future frames based on an initial observation and an action sequence, as in Figure 6. For evaluation, we use the held out BFA splits to test generalization to (1) unseen instances of objects within known categories (`Unseen Instances`) and (2) three entirely unseen object categories: donut, lime, and banana (`Unseen Categories`). We compare two variants of EAC-WM: EAC-WM-S, trained from scratch on BFA, and EAC-WM, which is first pretrained on three datasets (EgoDex, Agibot, and Droid), then finetuned on BFA. LAC-WM is pretrained then finetuned as in Section 4.3. All models take in raw robot actions, which is encoded into action embeddings through the explicit action encoders of EAC-WM or the action projector of LAC-WM. We report both image quality metrics (PSNR, FID, LPIPS) and a video quality metric (FVD), computed over 512 randomly sampled windows per split (see Table 1). Across all splits and metrics, LAC-WM performs best, followed by EAC-WM, while EAC-WM-S performs worst, highlighting the importance of pretraining and a unified action space for effective adaptation.

### 6.2 ROBOT PLANNING WITH ACTION SELECTION

We investigate the second research question of *whether conditioning a world model on a unified action space enhance cross-embodiment learning and downstream robot task planning performance compared to using a disjoint latent space*, especially when adapted to an unseen embodiment, by using the world model to select actions for completing a robot task.

**Problem Setup** As in Section 4.3, we consider the task of lifting an object from a kitchen counter and placing it back using the right arm of the BFA setup. We evaluate generalization to both `Unseen Instances` and `Unseen Categories`. We consider 22 unseen instances from 22 seen categories and 3 unseen categories. For each evaluation episode, we assume access to one demonstration trajectory for sampling a subgoal image every $p$ timesteps, resulting in a sequence of subgoal sets the robot must achieve to complete the task. For each subgoal, we sample $N$ candidate action sequences of length $p$. Given the current observation, the robot uses the world model to roll out predictions of future observations conditioned on action embeddings from action projector encoding each action sequence. The action sequence whose final predicted image is closest to the subgoal image is selected for execution, based on $l_2$ distance between the predicted final image embedding

| Val Set | Model | $\delta f \downarrow$ | $\delta f_g \downarrow$ | S.R.C ↑ | S.R.L ↑ | S.R. ↑ |
|---|---|---|---|---|---|---|
| Unseen Instances | VLA-mean | 26.49±0.06 | 27.07±0.02 | 0.84±0.04 | 0.56±0.00 | 0.20±0.03 |
| | VLA-random | 26.52±0.06 | 27.23±0.12 | 0.82±0.03 | 0.55±0.03 | 0.18±0.03 |
| | EAC-WM-S | 26.74±0.34 | 27.38±0.35 | 0.84±0.02 | 0.58±0.04 | 0.19±0.04 |
| | EAC-WM | 26.79±0.26 | 27.37±0.27 | 0.84±0.01 | 0.56±0.04 | 0.16±0.04 |
| | LAC-WM | **26.29**±0.15 | **26.87**±0.13 | **0.86**±0.03 | **0.60**±0.03 | **0.25**±0.03 |
| Unseen Categories | VLA-mean | 26.94±0.09 | 27.86±0.17 | 0.84±0.04 | 0.49±0.04 | 0.14±0.03 |
| | VLA-random | 26.93±0.06 | 27.77±0.07 | 0.80±0.01 | 0.48±0.01 | 0.12±0.02 |
| | EAC-WM-S | 27.16±0.39 | 28.00±0.41 | 0.84±0.02 | 0.49±0.07 | 0.17±0.03 |
| | EAC-WM | 27.19±0.28 | 28.03±0.27 | 0.82±0.01 | 0.49±0.04 | 0.14±0.04 |
| | LAC-WM | **26.73**±0.09 | **27.58**±0.11 | **0.85**±0.02 | **0.56**±0.02 | **0.18**±0.02 |
| Average | VLA-mean | 26.72±0.08 | 27.46±0.09 | 0.84±0.04 | 0.53±0.02 | 0.17±0.03 |
| | VLA-random | 26.72±0.06 | 27.50±0.09 | 0.81±0.02 | 0.52±0.02 | 0.15±0.02 |
| | EAC-WM-S | 26.95±0.37 | 27.69±0.38 | 0.84±0.02 | 0.54±0.05 | 0.18±0.03 |
| | EAC-WM | 26.99±0.27 | 27.70±0.27 | 0.83±0.01 | 0.52±0.04 | 0.15±0.04 |
| | LAC-WM | **26.50**±0.12 | **27.22**±0.12 | **0.85**±0.03 | **0.58**±0.02 | **0.22**±0.02 |

**Table 2: Robot planning performance using action sequences sampled from a VLA**. We compare the task performance of a VLA with action selection using a world model (EAC-WM-S, EAC-WM, LAC-WM) versus without action selection (VLA-mean, VLA-random). Results are averaged over 3 random seeds.

and the subgoal image embedding. We then move on to the next subgoal, repeating this process until the episode ends. An example of this rollout selection process is illustrated in Figure 7. In practice, to mitigate noise in the image embedding space, we select the top-$k$ action sequences with the closest final predicted images to the subgoal, and execute the average of these $k$ sequences.

**Metrics** Given the difficulty of this dexterous manipulation task, we evaluate performance using multiple metrics. Task Success Rate (S.R.) indicates whether the object was successfully lifted and placed back on the counter by the end of the episode. We also measure intermediate progress: Success Rate of Contact (S.R.C.), which indicates if the robot contacted the object, and Success Rate of Lifting (S.R.L.), which indicates if the object was lifted without touching the counter. To assess how closely the robot follows the subgoals, we compute the embedding distance between each subgoal image and the corresponding achieved image at each rollout step. We report the average embedding distance across all subgoals ($\delta f$), as well as the distance for the final goal ($\delta f_g$).

**Action Selection from VLA Samples** Instead of performing random action sequence sampling, which is inefficient for action optimization, especially for a difficult dexterous manipulation task, we explore sampling from alternative priors. With recent advancements in VLAs, we investigate whether world models can complement VLA by guiding action selection over VLA-generated candidates. Specifically, we leverage an off-the-shelf VLA model trained on the BFA dataset, which generates actions via a diffusion head conditioned on current observations, language instructions, and robot state. While we use the VLA with architecture specified in Section A.7, other VLA models with action sampling head can also be used. At test time, we generate diverse candidate actions by injecting different noise seeds into the flow-matching process, producing $N{=}500$ candidate sequences, each of length $p{=}100$. These sequences typically correspond to 4–5 subgoals per episode, resulting in a 20-step rollout for the world models. We use $k = 3$ to select the top 3 action sequences to calculate the average action sequence for execution, as described before.

We benchmark our world-model–guided action selection against two VLA-only baselines: **VLA-random**, which executes a single randomly sampled action sequence, and **VLA-mean**, which averages across all $N$ candidates. For action selection, we consider three world models: EAC-WM-S, EAC-WM, and LAC-WM. Each model is tested over 100 episodes per evaluation split, with 3 random seeds per experiment. An example rollout is shown in Figure 9. Results in Table 3 show that action selection guided by LAC-WM significantly outperforms the VLA-only baselines, particularly on the S.R.L and S.R metrics. Notably, LAC-WM achieves a **29.0% higher S.R.** than the strongest VLA baseline, a **22.2% improvement over EAC-WM-S**, and a **46.7% improvement over EAC-WM**, averaged across splits. While EAC-WM-S offers slight improvements over the VLA baselines, EAC-WM underperforms compared to both VLA-only methods and EAC-WM-S, indicating its limited ability to reliably predict long-horizon futures. This is likely due to its disjoint action space limiting

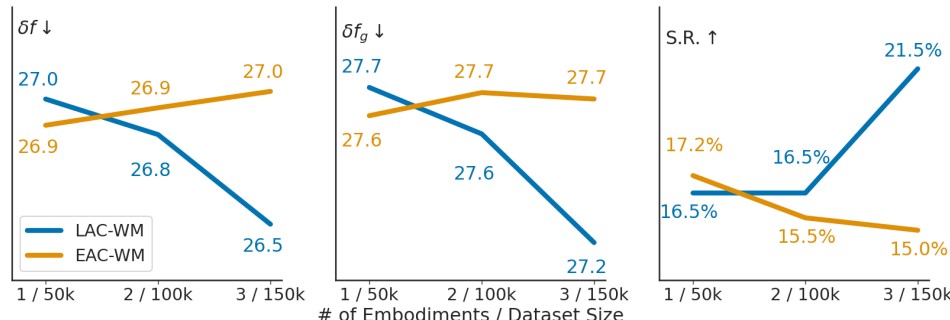

**Figure 4:** We show the image latent embedding distance $\delta f$ (left) and $\delta f_g$ (middle) and task success rate S.R. (right), for world models pre-trained with different number of embodiments. The x axis shows the number of embodiments and dataset size used in pre-training, from left to right of the x axis, the model is trained with Egodex only, Egodex and Agibot, and Egodex, Agibot and Droid dataset. The blue lines are the performance metrics corresponding to LAC-WM and the orange lines correspond to EAC-WM.

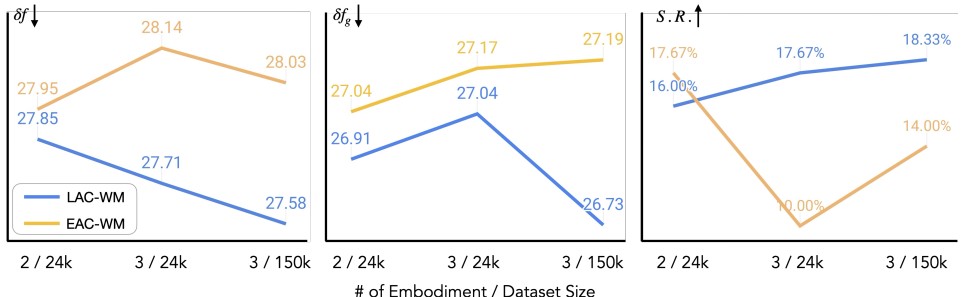

**Figure 5:** Experiments of scaling number of embodiments (first two points) or data samples (last two points).

cross-embodiment pretraining and downstream performance. This contrast highlights the advantage of LAC-WM 's unified action space in facilitating effective adaptation to unseen embodiments. Despite LAC-WM 's gains, overall S.R. remains low, reflecting the difficulty of this dexterous manipulation task. Additionally, we report the metric $\delta f$, which measures how effectively a method reaches the subgoal state, a lower $\delta f$ indicates better subgoal alignment. LAC-WM achieves lower $\delta f$ values than EAC-WM, suggesting more accurate subgoal reaching. However, since subgoal images are not used by VLA-only methods, they may attempt to reach different subgoal states. Therefore, both $\delta f$ and $\delta f_g$ serve as reference metrics rather than definitive performance indicators for those baselines.

**Scaling Law with Number of Embodiments and Data** We investigate the performance of world models as we scale the number of embodiments used in pre-training. Specifically, we pretrain both LAC-WM and EAC-WM on three subsets: (1) Egodex only, (2) Egodex and Agibot, and (3) Egodex, Agibot, and Droid. For each dataset, we use 50,000 samples, so models pretrained on more embodiments are exposed to proportionally larger amounts of data. Then we fine-tune the models on BFA and use them for VLA action selection. We report the S.R., $\delta f$ and $\delta f_g$ for each fine-tuned model. Results averaged over 3 seeds are shown in Figure 4. Interestingly, LAC-WM 's downstream planning performance improves as the number of pre-training embodiments increases, while EAC-WM shows the opposite trend with decreased S.R. and increased or nearly unchanged $\delta f$ and $\delta f_g$. We hypothesize this is because as the number of embodiment increases, the action embedding space become more disjoint for EAC-WM, making the model harder to learn from cross embodiment datasets, and less effective at adapting to the new BFA embodiment. These results highlight that a unified action space is critical for cross-embodiment pre-training.

We further investigate how world model performance scales with either the number of embodiments or the amount of pretraining data, as shown in Figure 5. The first two points in the plot correspond to settings where the total number of samples is fixed while increasing the number of embodiments. The last two points correspond to fixed embodiments while increasing the dataset size. In both scenarios, LAC-WM exhibits consistent performance improvements as either embodiments or samples increase. In contrast, EAC-WM performance degrades when adding more embodiments, and only improves when additional data comes from the same embodiment. These results suggest that EAC-WM cannot

efficiently leverage heterogeneous action spaces, further highlighting the advantage of learning a unified latent action representation.

| Val Set | Model | $\delta f \downarrow$ | $\delta f_g \downarrow$ | S.R.C $\uparrow$ | S.R.L $\uparrow$ | **S.R.** $\uparrow$ |
|---------|-------|------------------------|--------------------------|------------------|------------------|---------------------|
| Unseen Categories | LAC-WM-MD-CA | 26.98 | 27.80 | 0.83 | 0.46 | 0.13 |
| | LAC-WM | **26.73** | **27.58** | **0.85** | **0.56** | **0.18** |

Table 3: **Ablation study of Motion Decoding loss and Cross-Augmentation Inputs**

**Ablation study of Motion Decoding and Cross-Augmentation Inputs** We analyze the importance of motion decoding and cross-augmentation inputs for downstream robot planning. We compare LAC-WM with LAC-WM-MD-CA, a variant trained without the motion decoding loss or cross-augmentation, on the validation set of Unseen Categories. As shown in Table 3 (3 seeds), LAC-WM achieves a 38% higher success rate, suggesting that both motion decoding and cross-augmentation are crucial for learning latent actions that encode physically meaningful control information.

In summary, the superior performance of LAC-WM over EAC-WM indicates that *conditioning a world model on a unified action space enhances cross-embodiment learning and improves adaptation to unseen embodiments for robot action planning compared to using a disjoint latent space*.

## 7 CONCLUSION AND LIMITATIONS

In this work, we investigate foundational robot world model learning using a cross-embodiment dataset, focusing on two key questions proposed in Section 1. We compare LAC-WM, a world model conditioned on latent actions, with EAC-WM, which conditions on explicit action labels. Results demonstrate that latent action models yield more unified representations across embodiments, as evidenced by UMAP visualizations. Moreover, LAC-WM-guided planning significantly outperforms EAC-WM, achieving up to a 46.7% improvement in downstream task success rates when using VLA action samples. Importantly, we show that LAC-WM 's downstream performance scales positively with the number of embodiments used in pretraining, whereas EAC-WM exhibits the opposite trend. These findings highlight the effectiveness of a unified latent action space for addressing the unique challenges of cross-embodiment generalization and diverse action spaces in robotics.

Although the datasets used for pretraining include motion labels, this choice is deliberate: we aim to directly compare against a world model trained with explicit action supervision in order to assess the benefits of a unified latent action space. Prior work has shown the utility of latent actions in label-free video settings, but a direct comparison to label-conditioned models remains underexplored. Our auxiliary motion decoding loss encourages coherence in the latent action space without restricting the approach to labeled data, making it compatible with, and likely to benefit from, label-free video pretraining. The superior performance of latent action conditioning, even with motion labels, suggests that combining label-free video data and labeled cross-embodiment datasets during pretraining could further strengthen robot foundation models. Future work will explore incorporating large-scale label-free video data to enhance generalization and transfer capabilities in robotic world models.

Despite the significant performance improvement brought by LAC-WM, the absolute task success rate remains low. This indicates the difficulty of robot dexterous manipulation especially for unseen scenarios. Increasing the absolute task performance by designing better VLA policies and increasing the planning robustness remains an important future direction. A current limitation of our study is the absence of real-robot experiments. Although simulation allows extensive and controlled evaluation across diverse embodiments and objects, the reality gap poses additional challenges for practical deployment. Future work will incorporate real-world data collection and hardware validation to investigate the effectiveness of LAC-WM in real environments.

## REPRODUCIBILITY STATEMENT

The dataset, model architecture and training details have been described in Section 3, Section A.3 and Section A.5 for reproducibility. The RoboCasa simulation benchmark Nasiriany et al. (2024) is already open source. We commit to releasing all of the code, data, and models to accompany the paper.

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

# A    APPENDIX

## A.1    THE USE OF LARGE LANGUAGE MODELS (LLMS)

We used a large language model (ChatGPT) solely for writing refinement, including grammar correction and improving clarity and conciseness of the text. The model was not used for research ideation, experimental design, analysis, or content generation beyond language editing.

## A.2    PRETRAINING DATASET DETAILS

We conduct pretraining on three datasets: EgoDex (Hoque et al., 2025), Agibot (Bu et al., 2025a), and Droid (Khazatsky et al., 2025). The action space includes changes in end-effector poses and camera poses. The end-effector pose combines the wrist pose with additional dimensions for gripper or hand articulation. Each wrist pose is represented as a nine-dimensional vector containing three dimensions for translation and six for rotation. The camera pose follows the same nine-dimensional format. In Agibot, the end-effector pose has twenty dimensions total, ten per hand, with each consisting of the nine-dimensional wrist pose plus one dimension for the gripper state. In Droid, it is ten-dimensional for a single arm with the same structure. In EgoDex, the end-effector pose includes the nine-dimensional wrist pose plus sixty dimensions representing finger positions, corresponding to twenty keypoints in three-dimensional space, for a total of 138 dimensions for both hands. Camera motion is represented as a nine-dimensional change in pose for EgoDex and Agibot. For Droid, it is set to zero because the third-person camera remains fixed during each episode. The end-effector pose and camera motion vectors are concatenated as input to EAC-WM. For the motion decoder of LAC-WM, the latent action vector is divided evenly: the first half decodes the end-effector pose, and the second half decodes the camera pose.

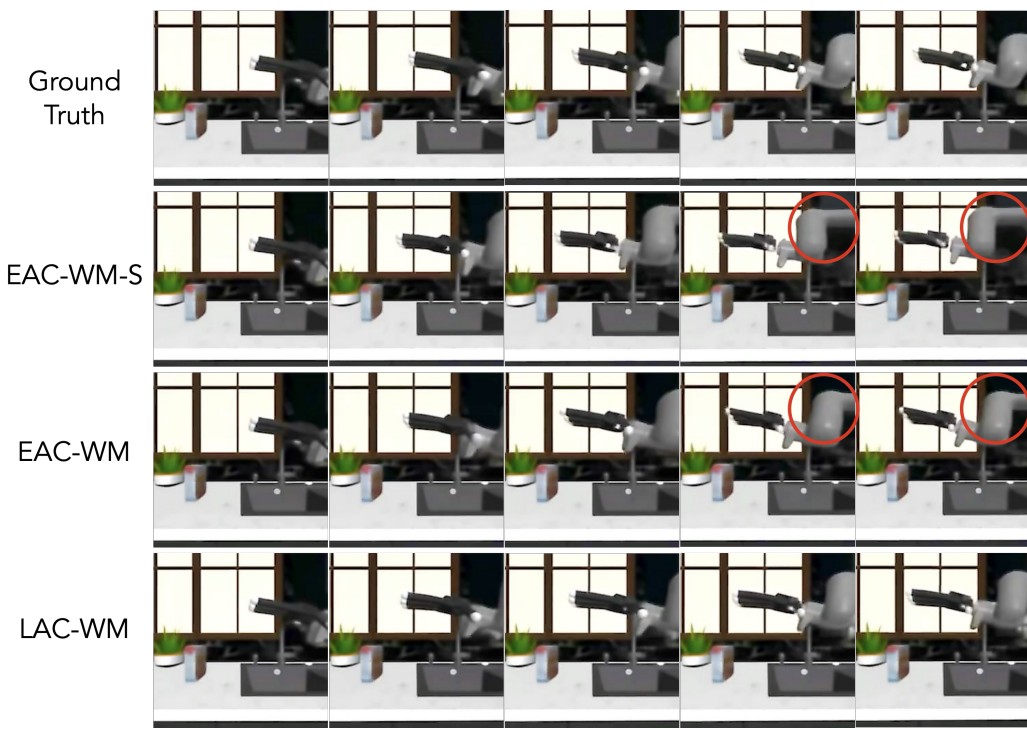

**Figure 6: Example of action conditioned imagined roll-out from the world models**. The top row displays the ground truth video, while the subsequent rows show action-conditioned imagined rollouts from EAC-WM-S, EAC-WM, and LAC-WM. While LAC-WM accurately predicts future observations, both EAC-WM-S and EAC-WM exhibit overshooting robot actions, highlighted by red circles, where the robot arm moves excessively. This suggests that LAC-WM better captures the robot dynamics.

## A.3 BFA Dataset

We use MimicGen (Mandlekar et al., 2023) along with RoboCasa simulator to produce large-scale synthetic pick and place datasets given 10 teleoperation trajectories. The teleoperators are instructed to pick up the object and place it either to the left or right of the initial position of the object with minimum of 0.1 meter movement. The data consists of 22 grasping object categories, resulting a total of 7,265 success and failure trajectories, with 20% of them being success trajectories. The robot consists of two franka arms mounted on the mobile platform. See Figure 6 for the examples of the dataset.

## A.4 Image decoder

To reconstruct pixels from V-JEPA2 Assran et al. (2025) embeddings, we train a custom Vision Transformer–based decoder. We first pretrain this decoder to reconstruct images from DROID and then finetune it on the BFA dataset. The decoder projects the input embeddings ($16 \times 16$ tokens, each of dimension 1408 for a $256 \times 256$ image) to a 1024-dimensional decoder space, followed by 12 self-attention layers with 32 attention heads each. The resulting features are then linearly mapped back to pixel space to reconstruct the output frames. We train the decoder using a standard mean-squared error (MSE) loss.

## A.5 Model Training and Inference

**Model Parameters** The details of our model parameters can be found in Table 4.

| Hyperparameter | Value |
|---|---|
| Image Tokenizer | V-JEPA 2 (1B) |
| Action Projector | 1,550,528 |
| Image Decoder | 153,845,952 |
| IDM: | |
| # Attention Blocks | 4 |
| # Attention Heads | 16 |
| # Latent Dimension | 512 |
| # Parameter | 46,939,968 |
| FDM: | |
| # Attention Blocks | 8 |
| # Attention Heads | 16 |
| # Latent Dimension | 512 |
| # Parameter | 93,844,480 |

**Table 4:** Hyperparameters for LAC-WM model architecture.

**Pretraining parameters** We pre-train both LAC-WM and EAC-WM for 80,000 iterations. For both models, we remove the temporal attention in the FDM for the first 60,000 iterations, to encourage strong conditioning on actions (Chen et al., 2024). The final 20,000 iterations fine-tune the model with additional temporal attentions in FDM using a temporal horizon of 8. The training costs around 4 days on 64 H200 GPUs.

**Model Inference** On an NVIDIA L40S GPU, the inference time is approximately 0.1 seconds for a batch size of 10 when predicting 8 future frames at 256×256 resolution.

## A.6 Action Selection from Training Dataset

To avoid potential bias introduced in VLA generated samples, we use action sequences sampled from the training dataset described in Section 4.3 to evaluate LAC-WM and EAC-WM under various fine-tuning strategies.

We set $N{=}500$, $p{=}30$, and $k{=}5$. This yields a sequence of 12-15 subgoals per episode. Since actions are chunked into 5-step segments for world model rollouts, this results in a 6-frame rollout per

Figure 7: **Examples of action selection rollout**. The leftmost image shows the initial observation, and the rightmost image depicts the goal state. Numbers in the top-left corner indicate time steps. In this example, three action sequences are sampled and rolled out by the world model, producing eight future predictions. The action sequence whose predicted final image has the smallest embedding distance ($\delta f_g$) to the goal image is selected for execution. Here, the action sequence in the top row is chosen.

| Val Set | Model | $\delta f \downarrow$ | $\delta f_g \downarrow$ | S.R.C↑ | S.R.L ↑ | S.R↑ |
|---------|-------|------------|-------------|--------|---------|------|
| Unseen Instance | EAC-WM-S | 25.00±0.04 | 26.53±0.06 | 0.59±0.03 | 0.22±0.02 | 0.11±0.02 |
| | EAC-WM-TFT | 25.05±0.01 | 26.49±0.10 | 0.54±0.03 | 0.15±0.06 | 0.09±0.02 |
| | EAC-WM | 25.03±0.04 | 26.53±0.12 | 0.61±0.03 | 0.21±0.02 | 0.11±0.01 |
| | LAC-WM-DFT | 24.82±0.08 | 26.32±0.20 | 0.61±0.03 | 0.19±0.02 | 0.07±0.01 |
| | LAC-WM-(2,3) | 24.88±0.04 | 26.51±0.08 | 0.54±0.09 | 0.21±0.01 | 0.09±0.01 |
| | LAC-WM-(1,2) | 24.90±0.09 | 26.37±0.12 | 0.59±0.07 | 0.21±0.05 | 0.09±0.06 |
| | LAC-WM | **24.60**±0.07 | **26.16**±0.19 | **0.65**±0.06 | **0.27**±0.06 | **0.12**±0.02 |
| Unseen Categories | EAC-WM-S | 25.75±0.04 | 27.46±0.07 | 0.56±0.02 | 0.08±0.03 | 0.03±0.03 |
| | EAC-WM-TFT | 25.68±0.06 | 27.39±0.11 | 0.54±0.00 | 0.13±0.05 | 0.07±0.01 |
| | EAC-WM | 25.68±0.15 | 27.50±0.09 | 0.54±0.07 | 0.16±0.04 | 0.08±0.02 |
| | LAC-WM-DFT | 25.57±0.04 | 27.35±0.05 | 0.53±0.08 | 0.19±0.01 | 0.09±0.02 |
| | LAC-WM-(2,3) | 25.78±0.18 | 27.48±0.28 | 0.51±0.09 | **0.21**±0.05 | 0.06±0.04 |
| | LAC-WM-(1,2) | 25.65±0.09 | 27.43±0.09 | 0.48±0.07 | 0.09±0.01 | 0.03±0.02 |
| | LAC-WM | **25.39**±0.02 | **27.21**±0.09 | **0.61**±0.01 | 0.18±0.03 | **0.10**±0.02 |
| Average | EAC-WM-S | 25.38±0.04 | 26.99±0.06 | 0.55±0.03 | 0.15±0.03 | 0.07±0.03 |
| | EAC-WM-TFT | 25.36±0.04 | 26.94±0.11 | 0.54±0.02 | 0.14±0.06 | 0.08±0.02 |
| | EAC-WM | 25.36±0.10 | 27.01±0.10 | 0.58±0.05 | 0.18±0.03 | 0.09±0.02 |
| | LAC-WM-DFT | 25.19±0.06 | 26.83±0.12 | 0.57±0.05 | 0.19±0.02 | 0.08±0.02 |
| | LAC-WM-(2,3) | 25.33±0.11 | 26.99±0.18 | 0.52±0.09 | 0.21±0.03 | 0.07±0.03 |
| | LAC-WM-(1,2) | 25.27±0.09 | 26.90±0.11 | 0.54±0.07 | 0.15±0.03 | 0.06±0.04 |
| | LAC-WM | **24.50**±0.04 | **26.68**±0.14 | **0.61**±0.03 | **0.22**±0.05 | **0.11**±0.02 |

**Table 5:** Robot planning performance for different methods using action selection from the training dataset.

sequence. We execute all 30 actions in the selected sequence. We evaluate each model on 50 episodes for both splits. Each episode is repeated three times using different random seeds, and we report the averaged results in Table 5. We compare the performance of EAC-WM-S, EAC-WM, and LAC-WM.

The results in Table 5 demonstrate that world-model–based action selection enables effective planning for dexterous manipulation. While the overall task success rate (S.R.) remains low due to the inherent difficulty of the task, models leveraging pretraining and fine-tuning (LAC-WM, EAC-WM) outperform the model trained from scratch (EAC-WM-S) on both evaluation splits. Notably, for the `Unseen Categories` split, LAC-WM achieves a final success rate of 10%, three times higher than EAC-WM-S. Pretraining also improves intermediate metrics: both the object contact rate (S.R.C.) and the object lifting rate (S.R.L.) increase, and the robot reaches states closer to the subgoal images, as indicated by lower image embedding distances ($\delta f$ and $\delta f_g$). Overall, LAC-WM achieves the best performance across all metrics, with significantly reduced embedding distances and higher task success rates. Figure 8 shows an example episode action execution, where LAC-WM successfully grasped and lifted the object while both EAC-WM-S and EAC-WM fails to lift the object, indicating LAC-WM is better at capturing fine-grained details of the robot hand posture and learning more accurate dynamics.

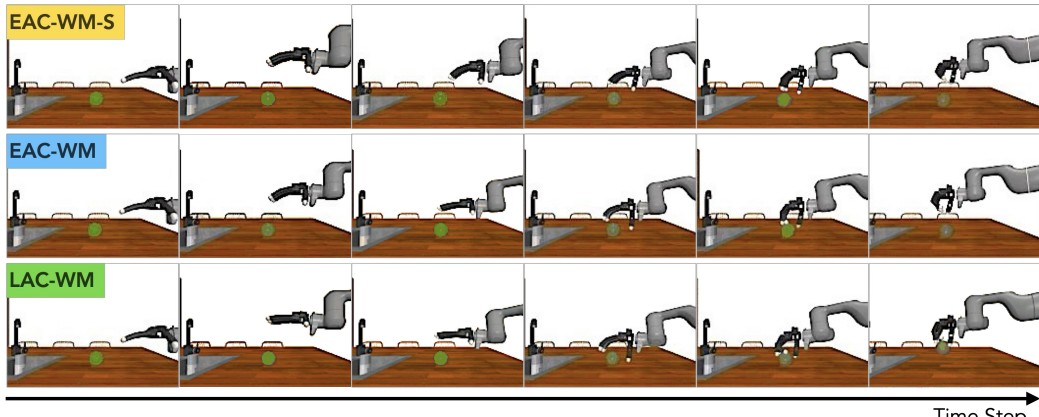

**Figure 8: Example rollout of robot action execution in the simulator**. From top to bottom shows the results from EAC-WM-S, EAC-WM, and LAC-WM. From left to right shows the progression over time. LAC-WM successfully grasps and lifts the object; EAC-WM makes contact but fails to complete the grasp; and EAC-WM-S misses the grasp entirely. This example illustrates LAC-WM's ability to capture fine-grained hand details and learn more accurate dynamics.

We further evaluate different fine-tuning strategies for both EAC-WM and LAC-WM, as summarized in Table 5. **EAC-WM-TFT** adopts a two-stage fine-tuning process: (1) fine-tuning the action encoder while freezing the FDM, followed by (2) end-to-end fine-tuning of both modules using LoRA (rank 2). **LAC-WM-DFT** performs end-to-end training of the action projector from scratch and fine-tuning of FDM using LoRA (rank 2). **LAC-WM-(2,3)** first trains an action projector from scratch with the FDM frozen, then jointly fine-tunes the action projector and FDM end-to-end using LoRA (rank 2). **LAC-WM-(1,2)** first fine-tunes the IDM and FDM of LAC-WM end-to-end using LoRA (rank 2), followed by training an action projector from scratch and end-to-end fine-tuning of the FDM using LoRA (rank 2). EAC-WM and LAC-WM are fine-tuned as in Section 4.3.

## A.7 VISION-LANGUAGE ACTION MODEL

Here we give details of the VLA model used in our experiments on world-model guided VLA action selection. The model processes multi-modal sensory inputs and generate action trajectories in a single forward pass. The architecture comprises the following key components:

- Visual Encoder: RGB images captured from the robot's onboard camera are processed using a pre-trained DINOv2 (Oquab et al., 2024) vision transformer. This encoder extracts high-level visual features, providing rich contextual information about the environment. The visual encoder is kept frozen during policy training.

- Language Encoder: Task instructions, provided as natural language commands or tokenized text, are embedded using a pretrained language encoder. This allows the policy to flexibly condition its behavior on diverse language inputs. The language embeddings are kept frozen during policy training.

- Proprioception and Action Encoders: The robot's proprioceptive states (end-effector positions and finger joint angles) are encoded using transformer-based encoders, trained from scratch. These encoders capture the robot's internal state, which is critical for precise control. The input dimension is 25: a 9-dimensional end-effector pose, and 16 finger joint angles.

- Multi-Modal Transformer Trunk: Encoded representations from all modalities (vision, language, proprioception, and previous actions) are fused using a 3-layer multi-modal transformer trunk. This component, trained from scratch, enables effective cross-modal integration and reasoning.

- Action Decoder (Diffusion Head): The fused multi-modal representation is decoded into action trajectories using a diffusion-based model head. Action space is future end-effector pose and finger joint angles, also 25 dimensional, predicted over an action horizon of 100. The diffusion head is trained with a diffusion loss, allowing the policy to model the distribution over possible future actions and generate diverse, high-quality action sequences.

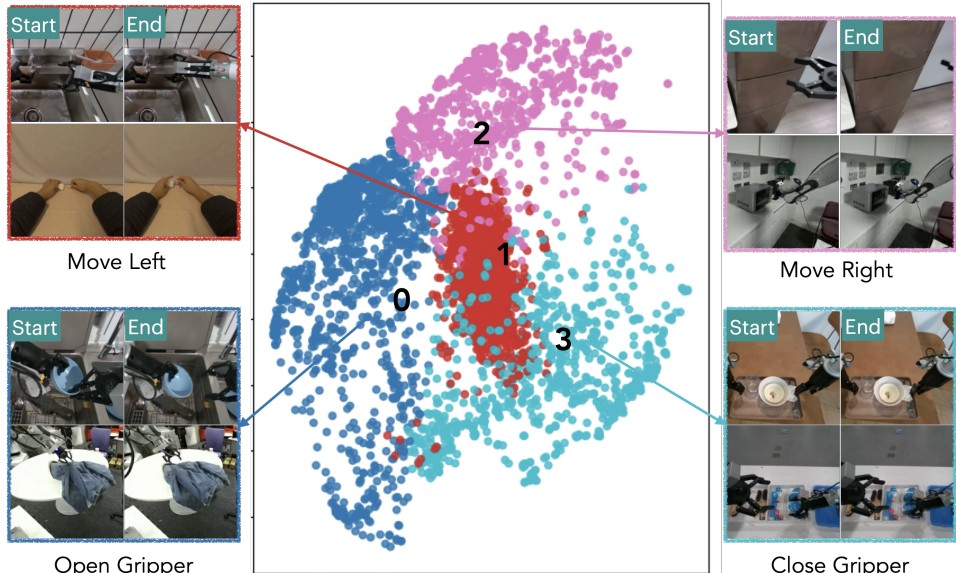

**Figure 9: Example roll-out of executing actions from a pre-trained VLA.** From top to bottom, the figure shows rollouts for: executing the mean of all sampled actions (VLA-mean), a random sampled action (VLA-random), and actions selected using EAC-WM-S, EAC-WM, and LAC-WM from the VLA samples. The task is to pick up the object and place it back on the counter. While LAC-WM successfully selects actions to complete the task, the other methods fail to grasp and lift the object.

**Figure 10: Latent Action PCA Analysis**. The representative examples from each of the cluster show that latent action in each cluster corresponds to semantically meaningful actions (e.g. opening and closing gripper, moving left and right) across different embodiments.

The policy predicts action sequences with a horizon length of 100 steps at a control frequency of 15Hz. We deploy the whole action chunk before replanning. The policy is trained with 2000 successful demonstrations of the robot picking and placing different objects.

## A.8 LATENT ACTION PCA ANALYSIS

To further study the physical meaning of the latent actions, we conduct PCA analysis on the latent action embeddings. Specifically, we obtain the PCA vectors on the latent action sequences (of length 4) extracted from all pretraining datasets (AgiBot, Droid and EgoDex). By projecting these high-dimensional latents onto the first two principal components, we visualize structure of the latent action space and perfrom K-means clustering (K=4) to identify distinct action clusters. For each

cluster, we select representative samples from each dataset that are closest to the cluster centroid. We then visualize the start and end images of the video sequence that produced these representative latent action sequences, as shown in Figure 10. The representative examples per cluster demonstrate that the learned latent actions capture consistent and meaningful motion patterns across different embodiments.

