# OpenReview forum: "Latent Action Robot Foundation World Models for Cross-Embodiment Adaptation"
_ICLR.cc/2026/Conference — Submitted to ICLR 2026_

### Official Review · Reviewer_aNtr · 2025-10-25

**Soundness:** 2
**Presentation:** 2
**Contribution:** 2
**Rating:** 4
**Confidence:** 2

**Summary:**

This paper introduces the Latent Action Conditioned Robot World Model to boost cross-embodiment learning, thus tackling the key challenges in today's robot learning field. However, the method itself is not novel at all, with every component having been seen in prior works. Also the experiment is poor, with only one "EAC-WM" baseline and a simple pick-and-place manipulation task. I do not think the draft is ready for ICLR 2026.

**Strengths:**

- The author highlights that the key challenge of robot learning is the heterogeneity of robotics data, which is crucial and important for building a general embodied intelligence.

**Weaknesses:**

- In the abstract, "LAC-WM" and "EAC-WM" are abrupt and lacking prior introduction.
- The first two paragraphs of the introduction lack references, while the challenges of cross-embodiment heterogeneity are well studied in many works, as listed in the references below.
- The baseline is inadequate, as only results of Explicit Action Conditioned World Model are presented.
- There is only one simple manipulation task which is also inadequate.

references:
- [1] Universal actions for enhanced embodied foundation models
- [2] Rdt-1b: a diffusion foundation model for bimanual manipulation
- [3] X-VLA: Soft-Prompted Transformer as Scalable Cross-Embodiment Vision-Language-Action Model
- [4] Scaling cross-embodied learning: One policy for manipulation, navigation, locomotion and aviation
- [5] Learning to Act Anywhere with Task-centric Latent Actions

**Questions:**

As mentioned in weaknesses above.

---

> ### Author Response · Authors · 2025-12-04
>
> We thank the reviewer for the feedback.
>
> ---
>
> ### **W1 & W2**
>
> We have added the full model name in the abstract and included the missing citations.
>
> ---
>
> ### **W3: Baseline adequacy**
>
> Our primary technical focus is on establishing a unified latent action space for multi-embodiment world-model learning. Therefore, the most relevant and representative baseline is the widely used Explicit Action Conditioned World Model (EAC-WM), which directly conditions on raw action spaces and is the dominant approach for handling heterogeneous robotic data in prior work.
>
> ---
>
> ### **W4: Limited robot experiments**
>
> While our primary evaluation focuses on a single dexterous manipulation task, we emphasize that this task is highly challenging: it requires long-horizon planning and multiple sequential skill executions (grasp, lift, transport, place) under distribution shift. To thoroughly evaluate generalization, we conduct experiments across 22 unseen object instances and 3 novel object categories, and run 100 episodes per condition over 3 random seeds, totaling **600 evaluation episodes**.
>
> Across all settings, LAC-WM consistently and significantly outperforms EAC-WM, achieving a **46.7% improvement** in planning success, demonstrating clear advantages of LAC-WM.
>
> To ensure strong empirical comparisons, we also evaluate both methods across multiple dimensions, including:
>
> 1. Latent space structure (UMAP)
> 2. Embodiment transfer experiments (human ↔ robot)
> 3. Rollout quality metrics (PSNR, LPIPS, FID, FVD)
> 4. Planning success across unseen objects
> 5. Scaling behavior with increasing embodiment diversity
>
> As pointed out by Reviewer 9u1V, our experiments are comprehensive and results consistently support the central hypothesis. Across all evaluation settings, LAC-WM consistently outperforms EAC-WM, demonstrating that learning a unified latent action representation provides benefits over explicit action conditioning.
>
> ---

---

### Official Review · Reviewer_Nfxz · 2025-10-25

**Soundness:** 3
**Presentation:** 3
**Contribution:** 3
**Rating:** 4
**Confidence:** 5

**Summary:**

This paper introduces a world model based on a latent action model, called **LAC-WM**. Recently, previous works have built a unified latent action space form both action and action-less videos by using an Inverse Dynamics Model and a Forward Dynamics Model. In this paper, the authors also introduce action decoding from latent actions to ground-truth actions to mitigate shortcut learning. Furthermore, to address the issue of feasibility--where the model cannot observe future frames during inference--the authors propose an action projector that encodes actions into latent actions. With these approaches, LAC-WM can imagine future scenarios even with unseen objects, and its planning capability outperforms the baselines, including directly using action labels.

**Strengths:**

- **Action-conditioned world model**
  - This paper introduces an action-conditioned world model based on a latent action approach. Most recent latent action-based approaches require future frames, transferring actions from source videos to predict future frames. However, LAC-WM introduces an action projector to align real actions with the latent action space, enabling the generation of action-conditioned future frames.
- **Planning with VLA**
  - Unlike discrete action spaces such as those in game environments, it is difficult to sample actions in continuous action spaces. To address this, the paper adopts a trained VLA for sampling. This approach appears to be a reasonable way to handle complex real-world tasks.

**Weaknesses:**

**Major Weakness**
- **Motion Decoder**
  - The authors claim that, to prevent shortcut learning where the latent action is encoded from future frames, they adopt a motion decoder. However, the provided analysis is insufficient. In Fig. 2, although the authors state that the visualization demonstrates a unified latent action space, even without the motion decoder, Agibot and Egodex already appear to be encoded in a unified action space. Moreover, even with the motion decoder, Droid and Egodex seem to form their own clusters in the center.
  - In LAOM [1] and CLAM [2], the models adopt an action decoder to improve robustness to distractors or to decode latent actions into real actions within the same environment. However, in this paper, the motion decoder is adopted to prevent shortcut learning. Since each motion has different properties -- Droid has one arm, Agibot has two arms, and Egodex has human hands -- it may introduce embodiment-specific information rather than effectively preventing shortcut learning. Based on Fig. 3, in terms of movements, there are no significant differences regardless of the motion decoder, and the results already show robustness to distractors, as noted in LAOM.
  - To better align the claimed benefit of the motion decoder with the authors' argument, further experiments such as ablation studies on robot experiments are required.
  - Beyond the motion decoder, the authors also argue that *cross-embodiment augmentation* mitigates the shortcut learning. However, it is difficult to understand how cross-embodiment augmentation achieves that effect without appropriate analyses.
- **Insufficient explanation**
  - The paper provides limited explanation of the proposed method and experimental setups.
  - For the pixel decoder, it is unclear how the embedding $x$ is decoded into the pixel space $I$. To the best of my knowledge, V-JEPA2 [3] does not provide an pixel decoder. Additionally, it is unclear whether the authors used a pretrained visual encoder from V-JEPA2 or trained it themselves.
  - In Tab 3, important details are missing for each setting, such as LAC-WM-DFT.
---

**Minor Weakness**
- **Robot experiments**
  - The provided experimental results are too weak to support the proposed method. Although the authors claim relative improvement, the gains are marginal. Moreover, the performance appears to depend heavily on the VLA.
  - Evaluating on a single task, especially one that is not a real-world task, is limiting.
- **Missing citations**
  - UniSkill [4] also adopts continuous latent actions from in-the-wild videos and demonstrates transferability across different embodiments.

---
[1] Nikulin, Alexander, et al. "Latent action learning requires supervision in the presence of distractors." arXiv preprint arXiv:2502.00379 (2025).

[2] Liang, Anthony, et al. "Clam: Continuous latent action models for robot learning from unlabeled demonstrations." arXiv preprint arXiv:2505.04999 (2025).

[3] Assran, Mido, et al. "V-jepa 2: Self-supervised video models enable understanding, prediction and planning." arXiv preprint arXiv:250

[4] Kim, Hanjung, et al. "UniSkill: Imitating Human Videos via Cross-Embodiment Skill Representations." arXiv preprint arXiv:2505.08787 (2025).

**Questions:**

**Major Questions**
- What is the benefit of using cross-embodiment augmentation and the motion decoder? Is there any in-depth analysis regarding how they mitigate shortcut learning?
  - Does each component affect downstream performance?
- If you fine-tune the IDM and FDM or downstream tasks, can the latent action space remain unified?
  - During fine-tuning, both the IDM and FDM are trained. During pretraining, by using various datasets, a unified latent action space is constructed. However, during fine-tuning, every module is updated, so I wonder whether the unified latent action space becomes biased toward the fine-tuned dataset.
  - When using the action projector to encode latent actions, can these be decoded using the motion decoder?
---
**Minor Questions**
- What are the implementation details? Which vision encoder is used (an open-weight model or manually trained one), and how is the embedding decoded into pixels?
- What is the meaning of each setting or row in Tab 3?
- If you plot latent actions from the same motion (or task) across different embodiments, are they well clustered? Additionally, are they distinguishable from different tasks within the same embodiment?
- Are there any real-world robot experiments?
- Are there any multi-task robot experiments?

---

> ### Author Response · Authors · 2025-12-04
>
> We appreciate the reviewer for the feedback and detailed comments.
>
> ---
>
> ## **Major Weakness**
>
> ### **Role of motion decoder and cross-augmentation in preventing shortcut learning**
>
> To directly address this concern, we conducted a new ablation study where both components are removed. As reported in **Table 3**, we compare LAC-WM with LAC-WM-MD-CA, a variant trained without motion decoding loss and without cross-augmentation, on the Unseen Categories validation set. LAC-WM achieves a **38% higher success rate**, demonstrating that both components are essential for ensuring physically meaningful latent actions rather than perceptual shortcutting.
>
> Below we provide detailed clarification to the reviewer’s specific points:
>
> ---
>
> #### **Importance of the motion decoder**
>
> We thank the reviewer for the detailed observations. Our definition of “shortcut learning” refers to the latent action embedding exploiting **purely perceptual cues** (e.g., optical flow), instead of representing **physically grounded motor actions**. The motion decoder provides explicit action supervision to avoid this.
>
> As shown in Fig. 2 (middle column), without motion supervision, Agibot and Egodex cluster together and remain separated from Droid. These two embodiments share similar egocentric visual perspectives, whereas Droid has varying third-person viewpoints. This suggests that the latent space aligns based on viewpoint similarity rather than shared action semantics. This is further validated by the action transfer results (Fig. 3). Although the transferred motions appear directionally correct, we observe floating hands and object shifts, which are visual artifacts consistent with optical-flow-like encoding. This shows that the behavior emerges from perceptual shortcutting rather than meaningful action grounding.
>
> In contrast, with a motion decoder, the latent action embeddings of all three embodiments are in a unified space (Figure 2). As the reviewer noted, small cluster offsets exist due to embodiment-specific kinematics (e.g., 1-arm vs. 2-arm vs. hand morphology), which we believe is an expected and desirable property that the latent action encodes both shared action intent and embodiment-specific execution constraints.
>
> In short, motion decoding is essential not for distractor robustness (as in LAOM/CLAM), but to enforce physically grounded control representations.
>
> ---
>
> #### **Effectiveness of cross-embodiment augmentation**
>
> As described in the paper, we adopt the cross-augmentation strategy from Igor[1], passing differently augmented observations to IDM and FDM. Since the method follows existing work, we do not claim novelty here and defer additional full analysis to Igor[1].
>
> ---
> #### **Insufficient explanation of methods and experimental setup**
> We have added clarifications in:
> - **A.4**: V-JEPA-2 decoder training
> - **A.5**: training details and experimental setup
> - **A.6**: expanded explanation of Table 3 (now Table 5)
> ---
>
> ## **Minor Weakness**
>
> ### **Limited robot experiments**
>
> Although evaluation focuses on a single dexterous manipulation task, we emphasize that this task is highly challenging: it requires long-horizon planning and multiple sequential skill executions (grasp, lift, transport, place) under distribution shift. To thoroughly evaluate generalization, we conduct experiments across 22 unseen object instances and 3 novel object categories, and run 100 episodes per condition over 3 random seeds, totaling 600 evaluation episodes. Across all settings, LAC-WM consistently and significantly outperforms EAC-WM, achieving a 46.7% improvement in planning success, demonstrating clear and meaningful gains.
> We also include an alternative action-sampling strategy (Appendix A.6), under which LAC-WM still improves by 22.2%, confirming robustness across policies. We agree broader real-robot evaluations will enhance impact and plan to pursue them as key future work.
>
> ---
>
> ### **Missing citations**
>
> The missing citations have been added in the revised related work.
>
> ---
>
> ## **Major Questions**
>
> (1) See Weakness 1
>
> (2.1) We appreciate the reviewer for raising this concern. Indeed, fine-tuning could potentially bias the latent action space toward the downstream dataset. To mitigate this, we adopt low-rank adaptation (LoRA with rank 2), which constrains parameter updates and preserves the pretrained latent structure. This lightweight adaptation enables the model to specialize to the downstream task while maintaining alignment across embodiments learned during pretraining. Furthermore, the consistent improvement of LAC-WM in downstream planning success after fine-tuning, especially compared to EAC-WM trained from scratch, provides empirical evidence that the pretrained unified latent action space remains beneficial and does not collapse into an embodiment-specific representation.
>
> (2.2) The action projector encodes explicit robot actions into the latent action, which can be decoded back using the motion decoder.
>
> ---

---

> > ### Author Response · Authors · 2025-12-04
> >
> > ## **Minor Questions**
> >
> > (1) We have added the implementation details in A.4 and A.5.
> >
> > (2) We have added explanations of Table 5 (originally Table 3) in A.6.
> >
> > (3) We have added the requested analysis in A.8. Specifically, we perform PCA on the latent action sequences and apply K-means clustering (K=4) to identify distinct groups of latent actions. For each cluster, we visualize the initial and final frames of representative video sequences in Figure 10, enabling interpretation of the associated physical motions. The results show that each latent action cluster corresponds to a semantically meaningful action primitive shared across embodiments, for example, opening/closing the gripper or left/right motion of the end-effector.
> >
> > (4) We acknowledge real world experiments are important future directions.
> >
> > (5)The current experiments are already multi-task as it involves grasping and transporting different objects.
> >
> > [1] Xiaoyu Chen, Junliang Guo, Tianyu He, Chuheng Zhang, Pushi Zhang, Derek Cathera Yang, Li Zhao, and Jiang Bian.
> > **Igor: Image-goal representations are the atomic control units for foundation models in embodied AI**, 2024.
> >
> > ---

---

### Official Review · Reviewer_9u1V · 2025-10-27

**Soundness:** 3
**Presentation:** 4
**Contribution:** 3
**Rating:** 6
**Confidence:** 3

**Summary:**

This paper proposes LAC-WM, a Latent Action-Conditioned World Model for robotics, aiming to build a unified latent action space shared across heterogeneous robot embodiments (e.g., human, humanoid, single-arm robots). The model consists of an inverse dynamics model (IDM), a forward dynamics model (FDM), a motion decoder, and an action projector. By pretraining on multiple embodiments (EgoDex, Agibot, Droid) and fine-tuning on an unseen robot (BFA), the authors show that this unified latent space leads to superior cross-embodiment adaptation and higher-quality video rollouts compared to an explicit-action baseline (EAC-WM). Quantitatively, LAC-WM improves planning success rate by up to 46.7% and scales positively with the number of pretraining embodiments.

**Strengths:**

1. **Clear motivation and well-scoped problem.**
The paper addresses a fundamental issue in robotic foundation models—heterogeneous action spaces across embodiments—which limits generalization. The proposed unified latent action space directly targets this challenge.
2. **Comprehensive experimental evaluation.**
The paper conducts extensive experiments, including (a) latent space visualization (UMAP), (b) cross-embodiment transfer (human ↔ robot), (c) quantitative rollout metrics (PSNR/FVD), (d) planning success rate, and (e) scaling analysis with embodiment number. Results consistently support the central hypothesis.

**Weaknesses:**

1. **Low absolute task performance.**
Despite relative improvements, the absolute task success rate remains modest (≈0.22). This indicates limited planning robustness, which should be acknowledged.

2. **Computational efficiency and scalability not analyzed.**
The paper does not report training cost, model size, or inference latency—important aspects for foundation model claims.

**Questions:**

1. Please report results when (i) fixing the total pretraining samples while increasing the number of embodiments, and (ii) fixing the number of embodiments while scaling the total samples. Does LAC-WM still show monotonic gains in UMAP alignment, rollout metrics, and planning success? Any signs of saturation?
2. Have the authors attempted any real-robot experiments or domain randomization studies to evaluate sim-to-real robustness?

---

> ### Author Response · Authors · 2025-12-04
>
> We appreciate the reviewer for acknowledging the importance of our approach for addressing the fundamental challenge of heterogeneous action spaces across embodiments, and for **highlighting the comprehensiveness of our experimental evaluation**.
>
> ---
>
> ### **W1: Low absolute task performance**
>
> We acknowledge that the absolute task performance remains low. This is because the task is highly challenging: it requires long-horizon planning and multiple sequential skill executions (grasp, lift, transport, place). In addition, the policy is trained on only 2k trajectories and is required to generalize to unseen scenarios. Therefore, we also report intermediate task progress metrics, where the object grasping and lifting success rate (**S.R.L.** in Table 2) is around **58%**. We have already acknowledged that improving final task performance through better VLA design and increased planning robustness is an important future direction (see the final section of the paper).
>
> ---
>
> ### **W2: Missing training cost, model size, or inference latency**
>
> We have added these details in Appendix A.5. Specifically:
> - Training time: ~4 days on 128 GPUs
> - Model size: Table 4
> - Inference latency: **0.1s** to predict 8 frames for a batch size of 10 on an NVIDIA L40S GPU
>
> ---
>
> ### **Q1: More controlled scaling experiments**
>
> We appreciate the reviewer’s thoughtful suggestion. We have added controlled scaling results in **Figure 5**, where:
> - The first two points: **fixed data, increased embodiments**
> - The last two points: **fixed embodiments, increased data**
>
> Results show that:
> - LAC-WM consistently improves with more embodiments or more data
> - EAC-WM degrades when more embodiments are added, and only improves when more same-embodiment data is provided
>
> These findings suggest that EAC-WM struggles to leverage heterogeneous action spaces, while a unified latent action space enables better scaling with cross-embodiment data.
>
> ---
>
> ### **Q2: Sim-to-real robustness with domain randomization**
>
> In this work, all planning experiments are conducted in simulation to enable rigorous statistical evaluation across diverse evaluation settings with controlled environment consistency. While sim-to-real transfer is a highly promising direction, our focus here is on establishing a unified latent action space for scalable world-model pretraining. Achieving physical experiments would additionally require real-world data for both policy learning and world model fine-tuning. We plan to explore physical experiments in future work.
>
> ---

---

### Official Review · Reviewer_r6e7 · 2025-10-30

**Soundness:** 2
**Presentation:** 2
**Contribution:** 2
**Rating:** 2
**Confidence:** 4

**Summary:**

This paper proposes the training of a cross-embodiment latent action space for world models. Unlike conventional methods, which use raw actions as the conditioning input for future frame prediction (referred to as EAC in this paper), the proposed LAC method trains an Inverse Dynamics Model to encode latent actions. A Forward Model is then used to predict the future frame, conditioned on the latent action and the current frame. The framework is trained on a mixed dataset across several embodiments, with the aid of additional motion supervision, which serves a regularization-like role. The proposed training paradigm is shown to be effective in aligning the action space across different embodiments, thereby promoting cross-embodiment adaptation.

**Strengths:**

1. Using raw actions as the condition for world model generation faces significant challenges due to the strong heterogeneity present in robotics datasets. The heterogeneous action spaces often lead to semantic misalignment, which hinders effective world model training. The idea of utilizing a shared latent action space to replace the original raw actions is promising, and this approach has been widely explored and proven effective in previous works.

2. The introduction of extra motion supervision is shown to be effective in accommodating the shared action space in the experiments presented in this paper. This insight suggests that identifying embodiment-agnostic supervision signals may be a valuable solution to bridge the gap between heterogeneous action spaces and facilitate alignment.

3. The paper presents some interesting and insightful behaviors of the LAC-WM through qualitative results. These observations provide valuable insights into the system's performance and behavior, highlighting the effectiveness of the proposed approach.

**Weaknesses:**

1. The proposed framework employs an auto-encoder style for training the action space without any explicit distribution constraint. This raises a key question about how actions are sampled from the obtained latent action space, especially since it is not trained as a distribution.

2. As this work aims to build a world model, more qualitative results are needed to demonstrate the quality of the generated frames. The paper could benefit from a deeper exploration of how the generated frames align with actual environmental states, providing a clearer assessment of the model’s effectiveness.

3. Using the distance of image embeddings as a metric to report or select executable actions is potentially unreliable, as it heavily depends on the choice of image encoder. If this approach is used, more ablation studies and analysis should be included to assess how to choose an appropriate image representation and to explore the sensitivity of the method to different encoder architectures.

4. There is a lack of sufficient analysis regarding the trained action space. The authors should provide more case studies or a detailed analysis of the physical meaning of each latent action. This would help clarify how well the latent space captures meaningful and interpretable robotic actions.

**Questions:**

Which model is used to obtain the image embeddings for the distance calculation in the experiments? Clarification on the choice of model and its impact on the results would be helpful, especially given the potential sensitivity of the metric to the image encoder.

---

> ### Author Response · Authors · 2025-12-04
>
> We thank the reviewer for the review and questions. We appreciate the reviewer for recognizing the importance of learning a shared latent action space to address the significant semantic misalignment challenges caused by heterogeneous robotic action spaces. We are also grateful that the reviewer acknowledged the effectiveness of the proposed approach and the valuable insights from our observations.
>
> ---
>
> ### **W1: Lack of explicit distribution, how are actions sampled?**
>
> We clarify that at inference time we do **not** sample actions from the latent space. Instead, we train an **action projector** that maps robot actions into the latent action space to condition the Forward Dynamics Model (FDM). Thus, the latent action serves purely as a **representation space**, and no distribution-based sampling mechanism is required.
>
> ---
>
> ### **W2: Alignment between generated frames and environment states**
>
> We appreciate this suggestion. We provide qualitative video examples in the supplementary, while Table 1 reports widely-used video generation metrics (PSNR, LPIPS, FID, FVD) and Table 2 demonstrates downstream control performance. Analyzing underlying environment states from predicted frames is an interesting but open research problem beyond the focus of this work. We believe the combination of perceptual evaluation and task success already provides a strong assessment of model effectiveness.
>
> ---
>
> ### **W3 / Q1: Sensitivity of image-embedding distance to encoder choice**
>
> We agree that the encoder choice may affect absolute values. To ensure consistency, we use the **same pretrained encoder (V-JEPA-2)** for both world-model training and evaluation. Furthermore, we conducted **600 total test episodes** across multiple seeds and two validation sets, where **LAC-WM consistently outperformed EAC-WM**, achieving an average **46.7% improvement**. These extensive results indicate the robustness and superiority of LAC-WM.
>
> ---
>
> ### **W4: Further analysis of latent action semantics**
>
> We have added a new analysis of the latent action space in Appendix A.8. Specifically, we perform PCA on the latent action sequences and apply K-means clustering (K=4) to identify distinct groups of latent actions. For each cluster, we visualize the initial and final frames of representative video sequences in Figure 10, enabling interpretation of the associated physical motions. The results show that each latent action cluster corresponds to a semantically meaningful action primitive shared across embodiments, for example, opening/closing the gripper or left/right motion of the end-effector. These findings further support that the latent action space captures interpretable and physically grounded control behaviors rather than encoding purely visual shortcuts.
>
>
> ---

---

### Author Response · Authors · 2025-12-04

## Response to Area Chair

We thank the Area Chair for overseeing our submission and the reviewers for their constructive feedback. In the revision, we added substantial new experiments, analyses, and clarifications that directly address the core concerns raised.

---

### 1. New Controlled Scaling Experiment

To verify that learning a unified latent action space enables better use of heterogeneous data, we added a controlled scaling study (Fig. 5):

- Increasing the number of embodiments while keeping total samples fixed
- Increasing total samples while keeping the embodiments fixed

Results demonstrate:

- LAC-WM consistently improves with more embodiments or more data
- EAC-WM degrades when new embodiments are introduced, and only improves when more same-embodiment data is provided

This directly addresses Reviewer 9u1V’s question on scalability and strengthens our central hypothesis.

---

### 2. Ablation on Motion Decoder and Cross-Embodiment Augmentation

To justify these components, we introduced a new double-ablation experiment (Table 3). We compare LAC-WM against a variant without motion decoding loss or cross-augmentation:

| Model Variant | Motion Decoder | Cross-Embodiment Augmentation | Success Rate (Unseen Categories) |
|--------------|----------------|-------------------------------|---------------------------------|
| LAC-WM       | Yes            | Yes                           | Highest (38% higher than ablation) |
| LAC-WM-MD-CA | No             | No                            | Significantly lower |

Conclusion:

- Motion decoding is essential for preventing perceptual shortcutting and enforcing physically grounded latent actions
- Cross-embodiment augmentation further strengthens embodiment alignment

This directly responds to major concerns from Reviewer Nfxz.

---

### 3. Latent Action Semantics: PCA and Clustering Analysis

We added an interpretability analysis of the latent space (App. A.8, Fig. 10):

- PCA projection on latent action trajectories
- K-means clustering (K = 4)
- Representative sequences visualized per cluster

Each cluster corresponds to a meaningful action primitive shared across embodiments, such as:

- Gripper open/close
- Left/right end-effector motion

This provides evidence that latent actions encode semantically grounded control behavior rather than visual shortcuts, addressing concerns from Reviewer r6e7 and Nfxz.

---

### 4. Expanded Implementation and Experimental Clarity

We added the missing technical details requested by reviewers:

- Full description of V-JEPA-2 decoder (App. A.4)
- Training cost, model size, and inference latency (App. A.5)
- Clear definitions of all ablation variants and table settings (App. A.6)
- Missing citations added to Introduction and Related Work

---

## Summary of Revisions

| Reviewer Concern | Revision Added |
|-----------------|----------------|
| Need evidence for scalability | Controlled scaling experiment |
| Why motion decoder and augmentation? | New double-ablation demonstrating 38% gain |
| Lack of action-space interpretability | PCA + K-means semantic analysis |
| Missing implementation details | Substantial clarifications in Appendices |
| Missing citations | Added comprehensive related work |

---

We hope the Area Chair will consider these major improvements and the clarified contributions in the revised decision.

---

### Meta-Review · Area_Chair_sWhv · 2025-12-05

**Summary:**

This paper introduces a robot world model, LAC-WM, which is constructed over a learned unified latent action space shared across diverse embodiments. The proposed LAC-WM method trains an Inverse Dynamics Model to encode latent actions. A Forward Model is then used to predict the future frame, conditioned on the latent action and the current frame.

Learning robot policies or world models in a unified latent action space is not a really new idea. Many existing studies have shown its effectiveness in cross-embodiment modeling. The paper received generally negative reviews, which might not be changed after the rebuttal process. Therefore, I recommend rejection.

**Reviewer Concerns:**

Some concerns that I think are still not well-addressed:
- Lack of more detailed discussion on many recent works (Reviewer aNtr)
- Lack of results to demonstrate the quality of generated frames, as in typical world models (Reviewer r6e7)
- Use of distance of image embeddings to select executable actions (Reviewer r6e7).
- Weak evaluation (Reviewer 9u1V, Nfxz, aNtr).


Some comments that I believe have been addressed or partly addressed during rebuttal, based on additional experimental results:
- The use of auto-encoder style training to extract action space without extra distribution constraint (Reviewer r6e7).
- The use of motion decoder (Reviewer Nfxz)
- Lack of analysis regarding the trained action space (Reviewer r6e7).
- Computation and scalability analyses (Reviewer 9u1V).
- Clarity issues (Reviewer Nfxz)

**Reviewer Scores:**

I think most of the reviewers are more likely to retain their scores unchanged.

---

### Decision · Program_Chairs · 2026-01-26

Reject